# Mechanosensitive Piezo Channels in Cancer: Focus on altered Calcium Signaling in Cancer Cells and in Tumor Progression

**DOI:** 10.3390/cancers12071780

**Published:** 2020-07-03

**Authors:** Dario De Felice, Alessandro Alaimo

**Affiliations:** Department of Cellular, Computational and Integrative Biology (CIBIO), University of Trento, 38123 Povo (Tn), Italy; dario.defelice@unitn.it

**Keywords:** piezo channels, cancer progression, calcium signaling, mechanotransduction

## Abstract

Mechanotransduction, the translation of mechanical stimuli into biological signals, is a crucial mechanism involved in the function of fundamentally all cell types. In many solid tumors, the malignant transformation is often associated with drastic changes in cell mechanical features. Extracellular matrix stiffness, invasive growth, and cell mobility are just a few hallmarks present in cancer cells that, by inducing mechanical stimuli, create positive feedbacks promoting cancer development. Among the molecular players involved in these pathophysiological processes, the mechanosensitive Ca^2+^-permeable Piezo channels have emerged as major transducers of mechanical stress into Ca^2+^ dependent signals. Piezo channels are overexpressed in several cancers, such as in breast, gastric, and bladder, whereas their downregulation has been described in other cancers. Still, the roles of mechanosensitive Piezos in cancer are somewhat puzzling. In this review, we summarize the current knowledge on the pathophysiological roles of these Ca^2+^-permeable channels, with special emphasis on their functional involvement in different cancer types progression.

## 1. Introduction

Virtually, all cell types are subjected to mechanical forces, comprising shear stress, stiffness, compressions, swellings, membrane curvatures, and tensional forces [1,2]. During evolution, cells have developed multiple mechanisms to sense and tackle external mechanical clues, modifying their size, shape, behavior and, eventually, remodeling the surrounding microenvironment [3,4]. The mechanism of conversion of physical stimuli into biochemical or electrical signals is known as mechanotransduction [5,6,7,8]. This mechanism plays a critical role in various cellular processes, such as development, proliferation, migration, and apoptosis, and it is directly involved in a wide range of physiological processes in mammals, including sensing of sound and touch, pain, blood pressure modulation, and urine flow [4,9,10,11].

Therefore, mechanotransduction is essential in homeostasis and physiological mechanisms, however, it can also promote the progression of disorders including cardiomyopathies, fibrosis, and cancer [12,13,14,15,16]. Of note, carcinogenesis is associated with alterations of mechanical properties of the affected tissue, which can magnify all the malignant behaviors including proliferation, migration, invasion, and angiogenesis [17,18,19,20]. In cancer, tumors tend to be stiffer than the adjacent normal tissue; for that reason, in soft tissues, like breast and abdomen, palpation is the initial diagnostic method to detect malignancies. Cancer cells within a dilated tumor are subjected to intense mechanical forces due to the increased extracellular matrix stiffness. Such forces could prompt their malignant progression via mechanosensitive structures present in the plasma membrane (see Section 3), which collaborate with cytoskeletal and focal adhesion complexes [21,22,23]. In both physiological and tumoral conditions, cells can sense, transduce, and transmit mechanical stimuli through a multitude of mechanosensitive molecules. Among all, focal adhesion complexes, integrins, cadherins, transcription factors (e.g., YAP/TAZ), G-protein coupled receptors, and mechanosensitive ion (MS) channels activate different signaling pathways [24,25,26,27]. In particular, MS cation channels, are the primary molecular players that are activated by mechanical forces giving rise to cellular signaling pathways involved in mechanotransduction mechanisms [28]. When activated, these cation channels are implicated in the Ca^2+^-signaling of mechanosensitive cells either directly, through a rapid Ca^2+^ influx from the extracellular space (e.g., TRP and Piezo channels), or indirectly, by preserving the electrochemical gradient needed for Ca^2+^ income (e.g., K_2P_, K_Ca_ channels) [28,29,30]. Over recent years, the function of some members of stretch-activated Ca^2+^-permeable TRP channels, and their involvement in malignancies, have started to emerge. Although it is an especially important topic, because of space limitations, we cannot expand this point further in this current review, but we encourage the readership to refer to these other recent reviews [29,31,32,33].

Overall, this review provides the current state of knowledge on the roles of Piezo channels into physiological contexts, along with their involvement in various cancer types progression, focusing in particular on Ca^2+^ signaling that depends on these channels.

## 2. Piezo Channels

In 2010, Patapoutian and colleagues discovered in a mouse neuroblastoma cell line a new gene, *Fam38A*, now known as *Piezo1*, which coded for a mammalian MS cation channel [34]. Further, a second Piezo channel was identified by sequence homology analysis and called *Piezo2* (*Fam38B*) [34]. The Piezo family of proteins is evolutionally conserved in animals, plants, and protozoa [34]. Over the past ten years, a large number of publications have established the importance of Piezo channels in cell mechanotransduction processes [35].

Like other MS ion channels, Piezo are pore-forming membrane proteins that are activated in response to mechanical stimuli applied to the cell membrane [36]. Specifically, Piezo channels are non-selective Ca^2+^-permeable channels whose gating can be stimulated by several mechanical stimuli affecting the plasma membrane, including compression, stretching, poking, shear stress, membrane tension, and suction (Figure 1) [34,37,38]. While other MS channels can be activated both by mechanical stretch and other physical and chemical stimuli, Piezo channels are the only ones to be primarily gated by mechanical stimuli [34,39,40]. Nevertheless, a recent study has demonstrated that Piezo channels show significant voltage sensitivity, thus they can also be viewed as important members of the voltage-gated ion channel family [41].

In mammalians, each monomer of these transmembrane proteins is extraordinarily large compared to other ion channels (over 2500 amino acids), with numerous (>18) predicted transmembrane domains. Structurally, they do not share sequence homology with any known ion channels and receptors [34,42]. Thanks to recent studies, mostly based on mouse Piezo1, many details of the structure and gating of Piezo channels have emerged. The recently resolved cryo-electron microscopy structures have clearly shown that both Piezo1 and Piezo2 are trimers organized around a cationic permeable pore [43,44,45,46,47].

Piezo1 and Piezo2 are largely expressed in different MS tissues. Piezo1 is abundant in mechanosensitive cells of the skin, lungs, bladder, intestines, and endothelial cells lining the lumen of blood vessels, whereas it is less expressed in colon, stomach, and kidney. Conversely, Piezo2 channels are highly expressed in dorsal root and trigeminal ganglia sensory neurons, lung, bladder, colon, and Merkel cells, where they primarily respond to touch and proprioception [34,39,40].

The physiological roles of Piezo channels are only beginning to be understood. In humans, Piezo1 is involved in sensing shear stress in endothelial cells and has crucial roles in the development and homeostasis of the circulatory system including formation of blood vessels, regulation of vascular tone and control of red blood cell (RBC) volume. RBC are cells that suffer significant mechanical forces while circulating in the bloodstream, and mechanosensitive Piezo1 channels act upstream of the calcium-activated potassium channel KCNN4, which regulates intracellular cationic content and cell volume [48]. Dalghi and colleagues recently discovered that Piezo1 is widely expressed in mouse urinary tract (kidneys, ureters, bladder, and urethra), thus suggesting a role as mechanotransductor sensible to wall tension and urine flow [49]. Furthermore, a recent study performed by Rosenblatt group demonstrated that, upon stretching or wounding, epithelia exhibit a fast proliferative response that permits the re-establishment of optimal cell density or sealing of the wound [50]. This effect was unambiguously dependent on Piezo1 function, as chemical or genetic inhibition of Piezo1 significantly abrogated the proliferative response. The authors proposed that this increased proliferation is modulated by Piezo1 and involves the extracellular signal regulated kinase 1 (ERK1), a calcium-activated kinase that play a role in controlling the G2/M transition [50]. Conversely, Piezo2 is mainly expressed in a subset of somatosensory neurons and its physiological role in mechanosensation comprises light-touch sensing, pain, and proprioception. Moreover, Piezo2 is expressed also in vagal and spinal sensory neurons that innervate the respiratory system, thus contributing to airway stretch sensing and lung inflation-induced apnea [51]. A recent study has shown that Piezo2 is expressed also in auditory hair cells that contain mechanosensitive channels in order to detect sound-induced vibrations. In this particular case, Piezo2 is localized in the apical membrane of hair cells and is responsible for a reverse-polarity current but not for the sensory-transduction current [52]. Finally, Piezo proteins have been proposed to play other critical roles in mechanotransduction processes, such as embryonic development, cell migration, and cell differentiation [38,53,54,55,56,57,58,59,60]. The central role they play in mammals is underscored by the fact that complete knockouts of Piezo1 or Piezo2 in mice result in lethal phenotype in utero [61].

In the last years, the crucial roles of Piezo channels in human physiology have been further supported by the identification of disease-causing mutations in both *Piezo* genes. Mutations and other Piezo1 abnormalities have been linked to several clinical disorders, such as generalized lymphatic dysplasia, dehydrated stomatocytosis, hereditary xerocytosis, heart failure, and diabetes mellitus [62,63,64,65,66,67]. In contrast, mutations in Piezo2 are associated with Gordon syndrome, Marden Walker syndrome, and scoliosis [68,69]. Finally, disruption of Piezo channels physiology by Piezo1 and/or Piezo2 aberrant expression is further linked to the onset of several tumors [29,31,39].

In the following section, we will describe the functional involvement of Piezo channels in different cancer types.

## 3. Piezo Channels in Cancer

For obvious reasons, Piezo channels have been primarily explored in cancer types originating from tissues subjected to a high degree of mechanical stress [22,31,70]. Piezos are stretch-activated Ca^2+^-permeable channels, therefore the deregulation of their expression levels, as occurs in many transformed cancer cells, critically affects calcium signaling pathways [71]. Changes in intracellular Ca^2+^ levels modulate several signaling pathways that control cellular processes, including those important for tumorigenesis and cancer progression [72]. Indeed, disruption of calcium homeostasis is involved in enhancing the characteristics of different hallmarks of cancer, such as apoptosis, cell migration, growth, invasion, and metastatization [73].

Upregulation of Piezo channels can be found in many malignancies, mostly of epithelial origin, and its known roles in cancer progression are reviewed in detail below and summarized in Table 1 at the end of Section 3.

### 3.1. Gastric Cancer

Piezo1 channels have largely been investigated for their involvement in gastric tumorigenesis [74,75]. In human gastric epithelial GES-1 cells, Piezo1 functions as a Trefoil factor family 1 (TFF1) binding protein, a protein specific to mucus-secreting cells, and this partnership was found to be essential for TFF1-mediated cell migration and invasion [74]. Piezo1 knockdown was associated with a downregulation of the integrin β1 expression that remove the migratory capacity of gastric cancer cells. Therefore, the data obtained by Yang and collaborators suggest that the complex Piezo1-TFF1 is required for regulating the expression of integrin β1 and contributes to the migration of gastric cancer cells [74].

Another piece of research pointed out the correlation among Piezo1 channels and gastric cell migration and tumorigenesis. The authors reported an overexpression of Piezo1 channels in most of the gastric cancer cell lines analyzed and in primary tumor samples compared with non-tumorous gastric tissues [75]. To support their results, the knockdown of Piezo1 in gastric cells was carried out, showing a decreased cell proliferation and invasion, and suppressing xenograft development in mice. Significantly, the authors demonstrated an enhanced sensitivity to Cisplatin or 5-FU following Piezo 1 siRNA transfection in some gastric cancer cell lines. Finally, Piezo1 knockdown induces morphological changes via GTP-Rac1 accumulation, suggesting that this channel is involved in the preservation of the cellular morphology by modulating the activity of Rho GTPase family members [75].

Overall, the results obtained in these papers suggest an oncogenic role for Piezo1 channels in human gastric cells, since they are required for cell proliferation, migration, and invasion to promote gastric cancer progression [74,75].

### 3.2. Breast Cancer

In the malignant human breast cancer cells MCF-7, a cellular model of primary invasive breast ductal carcinoma, the expression of Piezo1 is considerably increased compared to the MCF-10A cells, a model for normal mammary gland [76]. Piezo1 overexpression plays an important role in the migration of MCF-7 cells. The motility of breast cancer cells MCF-7 can be inhibited by tarantula toxin *Grammostola spatulata* mechanotoxin 4 (GsMTx-4), a non-specific blocker of Piezo channels. Lastly, the authors reported that breast cancer patients with high levels of Piezo1 mRNA in primary tumors have higher hazard ratios and shorter overall survival time, bringing to light the oncogenic role of Piezo1 in breast cancer, probably promoting migration, invasion, and metastatic dissemination [76].

Recently, the other member of the Piezo family, Piezo2, has been implicated in the modulation of breast cancer cell migration [77]. The expression of Piezo2 has been ascertained in normal and breast cancer tissues. Remarkably, Piezo2 is overexpressed in the in vitro cell model of triple negative breast cancer MDA-MB-231-BrM2 cells that are known to form metastasis in the brain [78]. Pardo-Pastor and collaborators elegantly demonstrated that the gating of Piezo2 generates Ca^2+^-influx which triggers downstream the RhoA-mDia pathway required for the regulation of actin cytoskeleton [77]. As the authors suggested, a possible mechanism for the activation of RhoA includes the mobilization of Fyn kinase and the activation of calpain. Finally, the results obtained by knocking down Piezo2 indicated that the mechano-activation of Piezo2 makes MDA-MB-231-BrM2 cells more susceptible with their ability to proliferate, invade, migrate and survive in the brain [77].

A recent study carried out by Lou et al. analyzed the expression, prognostic value, and underlying mechanisms of Piezo2 in breast cancer [79]. The authors found that Piezo2 was frequently downregulated in a large panel of breast cancer cell lines and clinical samples. Conversely, Piezo2 expression was found to be positively correlated with ER status but negatively correlated with triple-negative status in breast cancer, indicating that high expression of Piezo2 is strongly linked to progression of breast cancer. The data obtained also revealed a favorable prognosis of Piezo2 expression mainly in ER-positive or HER2-negative breast cancer, but a poor prognosis in patients with low expression of Piezo2, suggesting that low expression of Piezo2 might be a potential prognostic biomarker in breast cancer. Finally, it was demonstrated that Piezo2 expression may be targeted by five miRNAs and correlated with dysregulation of Hedgehog signaling pathway. Among the genes of this pathway, cell adhesion molecule-related/downregulated by oncogenes (CDON) was downregulated in breast cancer and the decreased expression of CDON indicated a poor prognosis. Moreover, it was also demonstrated that CDON expression was considerably decreased after knockdown of Piezo2, suggesting that CDON acts as a downstream of the channel. Overall, this study provides several evidences that Piezo2 downregulation might promote survival and progression of breast cancer [79].

In conclusion, further research will reveal if Piezo channels are linked with the tumorigenesis and progression of breast cancer, representing thus a potential therapeutic target for this type of malignancy.

### 3.3. Prostate Cancer

Several studies have demonstrated that intraparietal tension is considerably increased in human prostate cancer (PCa) compared with normal tissues and, significantly, this pressure may confer resistance to programmed cell death [80,81,82].

On this basis, Han and collaborators investigated the possible pathologic roles of Piezo channels in PCa development [83]. In this study, Piezo1 channel was demonstrated to be overexpressed in DU 145 and PC-3 human metastatic PCa cell lines. Furthermore, the expression of Piezo1 is largely elevated, both at mRNA and protein levels, in human prostate tumor samples compared to non-malignant tissues. Importantly, the knockdown of Piezo1 channel decreased cell growth and migration in vitro and suppressed the growth of prostate tumors injected in nude mice [83]. Based on their findings, the authors proposed the following tumorigenic mechanism (Figure 2).

The overexpression and the mechanical activation of Piezo1 in PCa cells may trigger a rise in Ca^2+^ intake. The increased level of intracellular Ca^2+^ may activate, via calmodulin (CaM) or CaM-dependent protein kinase II (CaMKII), the Akt/mTOR signaling pathways and, consequently, accelerate the cell cycle progression supporting cell proliferation, migration, and tumor growth. The suggested mechanism is reinforced by the fact that the downregulation of Piezo1 significantly decreases intracellular Ca^2+^ increments, preventing the phosphorylation of Akt and mTOR and arresting the cell cycle of PCa cells (Figure 2) [83].

Overall, this report points out a pivotal role of Piezo1 channel in human prostate cancer progression, even if further research is needed to determine its real involvement in PCa.

### 3.4. Glioma

Glioma is the most common form of brain neoplasm that originates from glial cells [84]. Recently, Piezo2 has been identified as a critical player in the progression of glioma, supporting cancer cell behavior and angiogenesis [85]. After subcutaneous injection of Piezo2-knocked down glioma cells in nude mice, Yang et al. reported a significant increase of apoptosis along with a reduced tumor cell proliferation and angiogenesis. Furthermore, in vitro experiments confirmed that Piezo2 down-regulation in endothelial cells resulted in the inhibition of glioma tumor cell growth, migration, and invasion. Based on the results obtained, the authors indicated that the tumorigenic effect of Piezo2 is related to Ca^2+^-dependent overexpression of Wnt11 and its secretion by endothelial cells which, in turn, boosted their angiogenic potential via β-catenin-dependent signaling [85]. Summarizing, this study supports pieces of evidence for Piezo2 as a crucial regulator of tumor angiogenesis in glioma.

In another elegant study, Chen and colleagues used *Drosophila* models of glioma to establish the roles of Piezo channels in tumors [86]. The results obtained in this study provided a clear picture in which the tumor stiffer microenvironment upregulates and activates Piezo1 to further enhance tissue mechanotransduction capacity, exacerbating glioma development and tumor cell proliferation. The authors postulated a potential mechanism where the mechanical activation of Piezo1 foments the Ca^2+^-dependent assembly of focal adhesions and triggers integrin-FAK signaling. In conclusion, based on the results obtained in this study, glioma cell stiffness initiates a feedforward process wherein Piezo1 over-activation leads to cancer progression (Figure 3) [86].

### 3.5. Osteosarcoma

Osteosarcoma (OS) is an aggressive bone neoplasia that affects primarily children and is characterized by a high tendency to metastasize in the lungs. OS is often associated with aberrant mechanical features [87], thus, it is reasonable considering that MS channels, especially Piezo channels, may play an essential role in OS tumorigenesis. In a study carried out by Jiang and colleagues, Piezo1 channels were found overexpressed in human OS cells [88]. The data of in vitro experiments indicated that Piezo1 could mediate the mechanical stress, inducing apoptosis and inhibiting cell proliferation of OS cells. In addition, further in vivo experiments demonstrated that the suppression of *Piezo1* gene expression enhanced tumor growth of such cells in immunocompromised mice. In brief, the results obtained in this study showed an oncosuppressive role for Piezo1 through Ca^2+^-induced apoptosis and the inhibition of the proliferative capability of OS cells [88].

Of note, Piezo1 is also upregulated in the Synovial sarcoma SW982 cell line [89]. In these cells, knocking down channel expression decreased the Ca^2+^ response to the agonist Yoda1 and cell migration, suggesting Piezo1 as a plausible cancer cell regulator.

### 3.6. Other Cancers

The activity of Piezo channels in lung cancers has been described as oncosuppressive [57,90]. Piezo1 mRNA levels are equally downregulated in small-cell lung carcinoma (SCLC) and in non-small-cell lung cancer (NSCLC) [91]. In NSCLC, knocking down both Piezo1 and Piezo2 by shRNA enhanced in vitro migratory capability and in vivo tumor growth [90]. Likewise, in SCLC the depletion of Piezo1 using siRNA resulted in increased cell migration and induced anchorage-independence growth in cancer cells [57], thus confirming the oncosuppressive role of Piezo channels in lung malignancies.

In the bladder, Piezo1 channel is found to be associated with Ca^2+^ influx and ATP release in urothelial cells. The Piezo1 channel is also involved in sensing mechanical stretch required for normal bladder function [53]. The pathological implications of Piezo1 and Piezo2 in bladder carcinoma in human and mice have been investigated by Etem and colleagues [92]. Both Piezo1 and Piezo2 mRNA transcripts in bladder cancer tissues were found to be substantially increased in comparison with normal bladder tissues. Additionally, immunohistochemical analysis of mouse bladder cancer cells revealed the overexpression of Piezo1 in the plasma membrane. The Piezo1 upregulation was directly correlated with tumor stage and size. Contrariwise, Piezo2 overexpression was found only in high-grade tumors.

Although the authors predicted a Piezo channel contribution in bladder carcinogenesis, the function(s) of these channels in the tumor progression has not been elucidated, thus, further investigation is required [92].

## 4. Piezo Channels as a Potential Therapeutic Target in Cancer

Taken together, the foregoing studies demonstrated the importance of mechanosensitive Piezo channels in pathophysiology, especially in cancer. They might be exploited as biomarkers for diagnostic purposes, as well as pharmacological and genetic targets for therapeutic strategies. Given that both Piezo1 and Piezo2 are upregulated in specific cancer cells and tissues, their potential use as tumor biomarkers for diagnosis and prognosis is indisputable. Before that, the scientific community should understand how those channels modulate cancer initiation and progression and their tissue specificity.

Besides, as discussed earlier, in some aggressive cancers Piezo1 is overexpressed, thus, tumor-selective inhibition of Piezo1 may be therapeutically efficient. Currently established blockers are gadolinium (Gd^3+^), ruthenium red [34,93] and the spider toxin GsMTx-4 [94,95], which are not only selective for Piezo1, but can inhibit other MS ion channels (Figure 1) [96]. Over the last years, three chemical activators of Piezo1, called Yoda1 [97], Jedi1, and Jedi2 [98] have been identified through high-throughput screening assays. These molecules regulate the mechanically induced response of Piezo1 channel by changing its activity and sensitivity. Most recently, an analog of Yoda1, known as Dooku1, was developed and proved able to reversibly antagonize Yoda1-induced activation of Piezo1 by competing for a specific channel binding site (Figure 1) [99]. The activation mediated by these agonists demonstrated the druggability of Piezo channels. Remarkably, these compounds have no effect on Piezo2. Overall, the potential drug targeting of Piezo1 in cancer therapy is still challenging and needs to be more thoroughly investigated.

Currently, two primary limitations preclude the pharmacological and genetic targeting of Piezo1 in the treatment of cancer. The first issue is the lack of specific inhibitors of Piezo1. However, the scenario might change soon as high-resolution cryo-EM structures of Piezo1 have recently been released [44,45,46,100]. This information would enable an optimized and rational design of molecule screening, which should accelerate the development of selective inhibitors of Piezo1. The second hurdle with blocking Piezo channels is their widespread expression in the body associated with their different physiological roles. For instance, siRNA and antisense oligonucleotide-based therapies should be evaluated for their ability to target selectively *Piezo* genes only in tumors [29]. In that regard, it has been reported that Piezo2 has 16 alternative splicing isoforms, which are expressed in mouse sensory neurons [101]. It would be interesting to investigate the expression of other tissue-specific isoforms and determine the detailed role of each one in mechanosensation, which might provide specific targets for pharmaceutical intervention.

## 5. Conclusions

The mechanosensitive Ca^2+^-permeable Piezo channels have emerged as primary transducers of mechanical stimuli into Ca^2+^ dependent signals. In this review, we briefly elucidated their important roles in human homeostasis and in pathophysiological conditions.

In recent years, our knowledge of the involvement of Piezo channels in cancer has considerably increased. Different types of cancer are characterized by an increased expression of both Piezo1 and Piezo2 whilst, in a few, they are downregulated, proof of their direct implication in these pathologies. Several studies in vitro and in vivo have demonstrated that the activation of Piezo channels can deliver local Ca^2+^ influx, thus, modulating key Ca^2+^-dependent signaling pathways associated with cancer cell migration, proliferation, and angiogenesis.

However, the roles of Piezo channels in cancer hallmarks are somewhat puzzling. To the best of our knowledge, it is likely that the activity of Piezos, as oncogenes or oncosuppressors, depends on the specific type of cancer and may involve various signaling pathways. Besides, not all results achieved in vitro or in vivo, using mainly mice, can be definitely translated to the large plethora of human cancers.

In conclusion, further basic research should uncover to what extent Piezos are involved in cancer progression and, exploiting the recently disclosed high-resolution molecular structure of these channels, be focused on the development of more selective drugs and innovative strategies for targeting Piezo channels in cancer treatment.

## Figures and Tables

**Figure 1 cancers-12-01780-f001:**
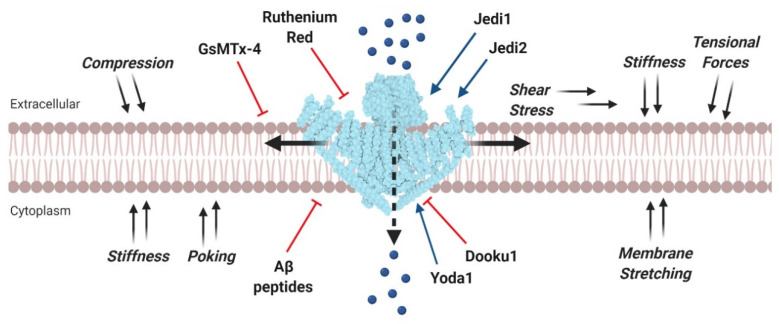
Mechanical and pharmacological modulation of Piezo1 channels. Mechanosensitive Piezo1 channels expressed on the plasma membrane are gated by various mechanical stimuli (black arrows). Channel activation allows a Ca^2+^ influx (blue spheres) into the cytoplasm which mediates countless cell responses. Piezo1 may be pharmacologically activated by agonists (blue arrows) or inhibited by channel pore blockers, competitive antagonists, and peptides, which distort the membrane mechanical properties (red inhibitory arrows). Structural data for Piezo1 channel were taken from Protein Data Bank (accession code: 3JAC) and created with BioRender.com.

**Figure 2 cancers-12-01780-f002:**
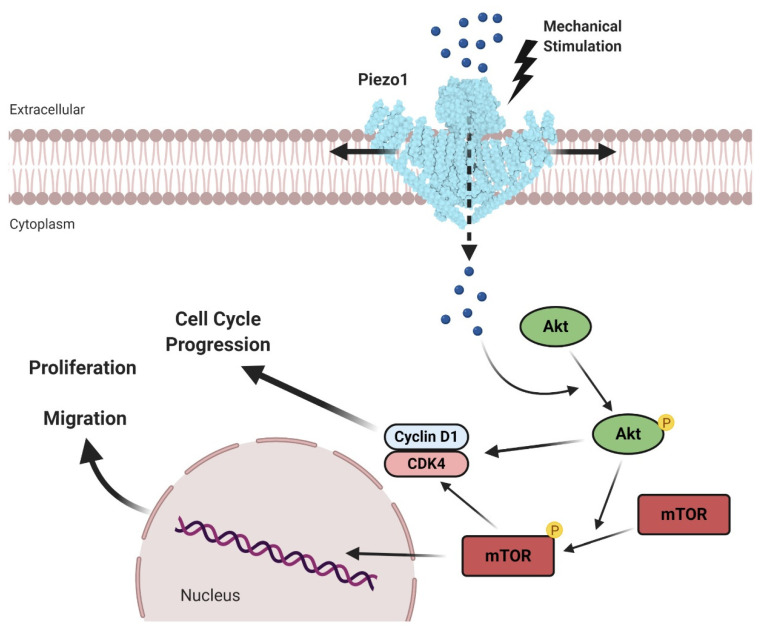
Piezo1 mediates the stretch-induced progression of prostate cancer. The mechanical gating of the overexpressed Piezo1 stimulates Ca^2+^ influx (blue spheres), which triggers the activation of both Akt and mTOR, finally upregulating the expression of the cell cycle key players cyclin D1 and CDK4. These intracellular signal transduction cascades may be responsible for cell cycle progression, cell proliferation, and migration, thus, may support the progression of prostate cancer. Adapted from reference [83]. Created with BioRender.com.

**Figure 3 cancers-12-01780-f003:**
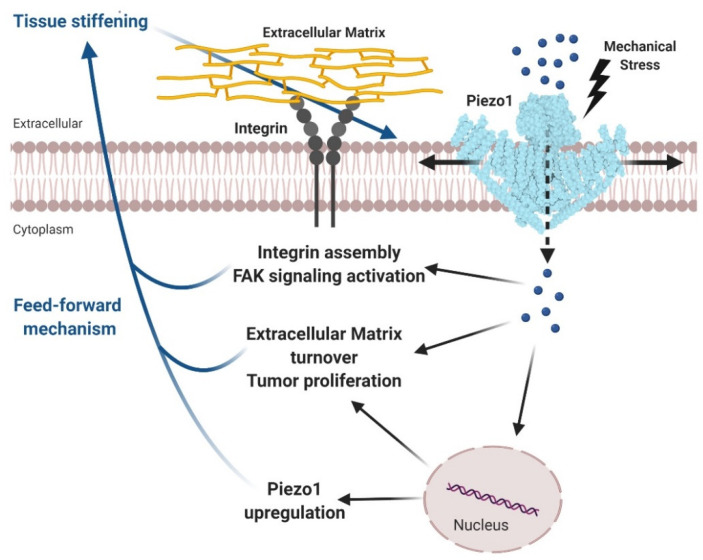
Tumor tissue stiffening activates Piezo1 promoting glioma aggressiveness. Tumor tissue stiffening represents the mechanical stimuli which activates Piezo1 facilitating Ca^2+^ influx (blue spheres) into cytoplasm. Consequently, intracellular Ca^2+^ increase, directly or indirectly, activates the integrin-focal adhesion signaling, stimulates cell proliferation and modulates the extracellular matrix remodeling. Besides, the enhanced expression of Piezo1 channels reinforces the mechanosensitivity of the tumor cells, creating a feedforward mechanism which leads to a further glioma progression. Adapted from reference [86]. Created with BioRender.com.

**Table 1 cancers-12-01780-t001:** Expression and roles of Piezo channels in cancer.

Cancer Type	Piezo Channel	Expression	Described Roles in Cancer	References
Gastric	Piezo1	Upregulation	Migration, Invasion, Proliferation	[74,75]
Breast	Piezo1 Piezo2	Upregulation Upregulation	Migration Migration, Invasion, Proliferation	[76] [77]
Prostate	Piezo1	Upregulation	Migration, Proliferation	[83]
Glioma	Piezo2 Piezo1	Upregulation Upregulation	Migration, Angiogenesis, Apoptosis resistance Proliferation	[85] [86]
Osteosarcoma	Piezo1	Upregulation	Oncosuppressor	[88]
Synovial Sarcoma	Piezo1	Upregulation	Migration	[89]
Lung	Piezo1, Piezo2	Downregulation	Oncosuppressor	[57,90,91]
Bladder	Piezo1, Piezo2	Upregulation	Unknown	[92]

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
