# Peer review of "Mechanosensitive Piezo Channels in Cancer: Focus on altered Calcium Signaling in Cancer Cells and in Tumor Progression"

_cancers, 2020, doi:10.3390/cancers12071780_

Round 1

Reviewer 1 Report

This review by De Felice and Alaimo entitled “Mechanosensitive Piezo channels in cancer: Focus on altered calcium signaling in cancer cells and in tumor progression” provides an interesting overview of current knowledge about Piezo channels roles in cancer. There have been very few reviews on this topic, making any new contribution to the field interesting. Overall, the manuscript is well written, with several illustrations clearly summarizing the text. I only have few minor comments:

  • On figures 2 and 3, “nucleus” should be clearly indicated.

  • Lines 58-59: the authors write that they will describe the physiological roles of Piezo channels. I believe that this part of the review could be expanded, in order to better present the numerous roles of Piezo channels. Indeed, while their main roles are listed, no explanation are provided, and the associated signaling pathways are never presented. A more comprehensive presentation of their physiological roles would help to better understand how they can promote cancer progression.

  • Lines 73-74: Contrary to the authors’ statement, Piezo channels have been proposed as prominently voltage-gated channels in a pathophysiological context (see Moroni et al, Nature Communication, 2018). Text should therefore be amended in order to reflect this possibility.

Author Response

Reviewer 1

This review by De Felice and Alaimo entitled “Mechanosensitive Piezo channels in cancer: Focus on altered calcium signaling in cancer cells and in tumor progression” provides an interesting overview of current knowledge about Piezo channels roles in cancer. There have been very few reviews on this topic, making any new contribution to the field interesting. Overall, the manuscript is well written, with several illustrations clearly summarizing the text. I only have few minor comments:

We thank reviewer 1 for his/her appreciation, valuable time, suggestions and constructive comments to further improve the quality of our manuscript

The reviewer can find the corrections/changes required in the revised text (in green)

On figures 2 and 3, “nucleus” should be clearly indicated.

Following this recommendation, we have included “nucleus” in Figure 2 and 3. Thanks you for the suggestion.

Lines 58-59: the authors write that they will describe the physiological roles of Piezo channels. I believe that this part of the review could be expanded, in order to better present the numerous roles of Piezo channels. Indeed, while their main roles are listed, no explanation are provided, and the associated signaling pathways are never presented. A more comprehensive presentation of their physiological roles would help to better understand how they can promote cancer progression.

We thank the referee for this constructive suggestion. As required, this section has been changed following the reviewer indications. We have referred to literatures and papers, re-analyzed the data and reconstructed this section to improve the quality of our paper. Obviously, this was beyond the scope of this review, however we think that this chapter is more complete now, and we thank again the referee for bringing this to our attention.

Lines 73-74: Contrary to the authors’ statement, Piezo channels have been proposed as prominently in a pathophysiological context (see Moroni et al, Nature Communication, 2018). Text should therefore be amended in order to reflect this possibility.

The referee is right. We have rephrased this section and added the suggested reference [41]. We thank the referee for this constructive suggestion.

Reviewer 2 Report

This is a very informative and interesting review about the role of mechanosensitive Piezo channels in cancer. There are a number of grammatical and spelling errors, together with omissions from the recent literature and some points that could be expanded on, as outlined below by line number. Otherwise, I think this is an excellent review. 

L40 tissue, comma should a semi-colon

L56 we cannot deepen this item in this manuscript. I would say we cannot expand this point further in this current review, but we encourage the readership to refer to these other recent reviews

L63 which encoded for should be which coded for

L68 As other should be Like other

L86 they primarily responds should be they primarily respond

L87 pharmacological activators are all selective for piezo 1 not piezo 2 as mentioned in L 264 to 272 so this should be made clearer in Fig. 1

L88 on plasma membrane should be on the plasma membrane

L99 play in mammalians should be play in mammals

L99 a recent paper about cell migration is not mentioned and should be: ‘Agonist-induced Piezo1 Activation Suppresses Migration of Transformed Fibroblasts’ PMID: 31029419

L105 Instead, mutations should be In contrast, mutations

L111 Unavoidably, I would say For obvious reasons,

L121 at the end of this chapter. Should be at the end of this review.

L123 their implication should be their involvement

L127 the integrin β1 expression remove the

L129 contribute to should be contributes to

L131 research work should be piece of research

L131-136 the authors should discuss the other major findings mentioned in this paper. Such as The effects of Piezo 1 channel on drug sensitivity (Cisplatin or 5‐FU) Zhang, J., Zhou, Y., Huang, T., Wu, F., Liu, L., Kwan, J. S., ... & Kang, W. (2018). PIEZO1 functions as a potential oncogene by promoting cell proliferation and migration in gastric carcinogenesis. Molecular carcinogenesis, 57(9), 1144-1155.

L131 gastric cells should be gastric cell

L133 gastric cancer cell lines put the before gastric

L140 it would be useful if the authors could be more specific in which type of breast cancer Piezo channels play a role.

L143 is strictly involved I would say plays an important role

L149-158 A recent paper on breast cancer is not mentioned linking down-regulation of Piezo2 to poor prognosis which is relevant to this discussion so should be cited: ‘Five miRNAs-mediated PIEZO2 Downregulation, Accompanied With Activation of Hedgehog Signaling Pathway, Predicts Poor Prognosis of Breast Cancer’ PMID: 31058608

L161-2 for this malignancy. I would say for this type of malignancy.

L175 the Ca2+ influx remove the

L186 cell cycle of PCa cell should be cell cycle of the PCa cell or cell cycle of PCa cells

L188 its real implication in PCa I would say its real involvement in PCa

L210 Drosophila should be in italics

L207 glioma cells stiffness should be glioma cell stiffness or glioma cells’ stiffness

L217 suppression of Piezo1 gene I would say suppression of Piezo1 gene expression

L217 enhanced tumor growth I think you need to add enhanced tumor growth of such cells

L220 in Synovial sarcoma should be in the Synovial sarcoma

L239 with the Ca2+ influx remove the

L241 implication should be implications

L247 was revealed I would say was found

L248 a Piezo channels should be a Piezo channel

L264 others MS ion channels should be other MS ion channels

L268 was delivered and proved to I would say was developed and proved able to

L272 requires to be deeply investigated. I would say needs to be more thoroughly investigated.

Author Response

Reviewer 2

This is a very informative and interesting review about the role of mechanosensitive Piezo channels in cancer. There are a number of grammatical and spelling errors, together with omissions from the recent literature and some points that could be expanded on, as outlined below by line number. Otherwise, I think this is an excellent review. 

We thank reviewer 2 for his/her appreciation, valuable time, suggestions and constructive comments to further improve the quality of our manuscript

The reviewer can find the corrections/changes required in the revised text (in red)

L40 tissue, comma should a semi-colon

Amended, Thank you

L56 we cannot deepen this item in this manuscript. I would say we cannot expand this point further in this current review, but we encourage the readership to refer to these other recent reviews

As required the sentence has been rephrased. Thank you for the suggestion

L63 which encoded for should be which coded for

Amended

L68 As other should be Like other

Amended

L86 they primarily responds should be they primarily respond

This typo has been corrected

L87 pharmacological activators are all selective for piezo 1 not piezo 2 as mentioned in L 264 to 272 so this should be made clearer in Fig. 1

As recommended, we have corrected this as suggested in the revised figure 1 and figure legend 1

L88 on plasma membrane should be on the plasma membrane

Thank you again. It has been amended

L99 play in mammalians should be play in mammals

This typo has been corrected

L99 a recent paper about cell migration is not mentioned and should be: ‘Agonist-induced Piezo1 Activation Suppresses Migration of Transformed Fibroblasts’ PMID: 31029419

As suggested, this reference has been added: Chubinskiy-Nadezhdin et al. [60]

L105 Instead, mutations should be In contrast, mutations

Amended

L111 Unavoidably, I would say For obvious reasons,

Thank you for the suggestion

L121 at the end of this chapter. Should be at the end of this review.

The table is at the end of chapter 3. Thank you

L123 their implication should be their involvement

Amended

L127 the integrin β1 expression remove the

Thank you again. It has been amended

L129 contribute to should be contributes to

Amended. Thank you

L131 research work should be piece of research

Amended. Thank you

L131-136 the authors should discuss the other major findings mentioned in this paper: PIEZO1 functions as a potential oncogene by promoting cell proliferation and migration in gastric carcinogenesis. Molecular carcinogenesis, 57(9), 1144-1155. Such as The effects of Piezo 1 channel on drug sensitivity (Cisplatin or 5‐FU) Zhang, J., Zhou, Y., Huang, T., Wu, F., Liu, L., Kwan, J. S., ... & Kang, W. (2018).

We thank the referee for this constructive suggestion. As required, this section has been changed following the reviewer indications.

L131 gastric cells should be gastric cell

Thank you. It has been amended

L133 gastric cancer cell lines put the before gastric

Thank you again. It has been amended

L140 it would be useful if the authors could be more specific in which type of breast cancer Piezo channels play a role.

Following this recommendation, we have specified the breast cancer models used in these studies. Thanks you for the suggestion.

L143 is strictly involved I would say plays an important role

Thank you for the suggestion

L149-158 A recent paper on breast cancer is not mentioned linking down-regulation of Piezo2 to poor prognosis which is relevant to this discussion so should be cited: ‘Five miRNAs-mediated PIEZO2 Downregulation, Accompanied With Activation of Hedgehog Signaling Pathway, Predicts Poor Prognosis of Breast Cancer’ PMID: 31058608

We thank the referee for this constructive suggestion. As required, this section has been changed following the reviewer indications. We have added the indicated reference, re-analyzed the data and reconstructed this section to improve the quality of our paper

L161-2 for this malignancy. I would say for this type of malignancy.

Amended. Thank you

L175 the Ca2+ influx remove the

This typo has been corrected

L186 cell cycle of PCa cell should be cell cycle of the PCa cell or cell cycle of PCa cells

Thanks again. This typo has been corrected

L188 its real implication in PCa I would say its real involvement in PCa

Thank you for the suggestion

L210 Drosophila should be in italics

Amended

L207 glioma cells stiffness should be glioma cell stiffness or glioma cells’ stiffness

Thank you for the suggestion

L217 suppression of Piezo1 gene I would say suppression of Piezo1 gene expression

Thank you again for the suggestion

L217 enhanced tumor growth I think you need to add enhanced tumor growth of such cells

Amended

L220 in Synovial sarcoma should be in the Synovial sarcoma

Amended

L239 with the Ca2+ influx remove the

This typo has been corrected

L241 implication should be implications

Amended

L247 was revealed I would say was found

As recommended, we have corrected this

L248 a Piezo channels should be a Piezo channel

This typo has been corrected

L264 others MS ion channels should be other MS ion channels

Amended

L268 was delivered and proved to I would say was developed and proved able to

As required the sentence has been rephrased. Thank you for the suggestion

L272 requires to be deeply investigated. I would say needs to be more thoroughly investigated.

Thank you for the suggestion